# Integrated Microbiota and Metabolomics Analysis of *Candida utilis* CU-3 Solid-State Fermentation Effects on Cottonseed Hull-Based Feed

**DOI:** 10.3390/microorganisms13061380

**Published:** 2025-06-13

**Authors:** Deli Dong, Yuanyuan Yan, Fan Yang, Huaibing Yao, Yang Li, Xin Huang, Maierhaba Aihemaiti, Faqiang Zhan, Min Hou, Weidong Cui

**Affiliations:** Xinjiang Laboratory of Special Environmental Microbiology/Xinjiang Uygur Autonomous Region Academy of Agricultural Sciences, Institute of Microbiology, Urumqi 830091, China; 18265390796@163.com (D.D.); yuanyuanyan0114@xaas.ac.cn (Y.Y.); yangfan312au@163.com (F.Y.); yaohuaibing@stu.xju.edu.cn (H.Y.); liyang@xaas.ac.cn (Y.L.); 15739456858@163.com (X.H.); malika511@126.com (M.A.); zfq_xj@126.com (F.Z.)

**Keywords:** untargeted metabolomics, fermented cottonseed hull, ITS sequencing, *Candida utilis*

## Abstract

Solid-state microbial fermentation (SSMF) has been established as an effective bioprocessing strategy to augment the nutritional value of plant-derived feed substrates while reducing anti-nutritional factors (ANFs). However, there have been limited studies on the effects of microbial solid-state fermentation on the nutritional value and potential functional components in cottonseed hulls. This study investigated the nutritional enhancement of cottonseed hulls through anaerobic solid-state fermentation mediated by *Candida utilis* CU-3, while exploring the functional potential of the fermented feed by analyzing fungal community dynamics and metabolite profiling. The laboratory-preserved free gossypol-degrading strain *Candida utilis* CU-3 was inoculated into unsterilized, crushed, and screened cottonseed hulls for solid-state fermentation at room temperature for 10 days. The results demonstrated that, compared to the control group, the experimental group achieved a 61.90% increase in free gossypol degradation rate, a 27.78% improvement in crude protein content, and a 5.07% reduction in crude fiber, while crude fat showed no significant difference. During the fermentation process, microbial diversity decreased, and *Candida utilis* CU-3 became the dominant species. Untargeted metabolomics data revealed that cottonseed hulls inoculated with *Candida utilis* CU-3 produced functional bioactive compounds during fermentation, including chrysin, myricetin (anti-inflammatory, antibacterial, and antioxidant activities), and ginsenoside Rh2 (anticancer, antibacterial, and neuroprotective properties). This study demonstrates that inoculating *Candida utilis* CU-3 into cottonseed hulls enhances their health-promoting potential through the biosynthesis of diverse functional metabolites, providing a theoretical foundation for improving the nutritional profile of cottonseed hull-fermented feed.

## 1. Introduction

With population growth and rising living standards, the consumption of animal products necessitates increased feed production to sustain livestock development [1]. Among the many feed resources, cotton, an important commercial crop, is widely grown in more than 70 countries worldwide [2]. During cotton processing, substantial byproducts are generated, including cottonseed hulls and cottonseed meal [3]. Cottonseed hulls have become a widely utilized regional feed ingredient feed in animal feed formulations [4]. However, cottonseed hulls contain significant amounts of free gossypol, an anti-nutritional factor that can be effectively removed through solid-state microbial fermentation (SSMF) [5].

Solid-state microbial fermentation (SSMF) has been recognized as a technically straightforward and effective methodology to enhance nutritional profiles while reducing anti-nutritional factors (ANFs) through microbial bioconversion processes [6]. Yao et al. significantly enhanced the nutritional quality of soybean meal through anaerobic solid-state fermentation using *Bacillus subtilis*, which notably increased the crude protein and acid-soluble protein contents in the fermented soybean meal [7]. The nutritional characteristics of fermented feed are closely related to the microbial composition of the feed [8]. Currently, the primary microorganisms employed in microbial solid-state fermentation (SSF) include *Lactobacillus* spp., *Bacillus* spp., and yeast, among others [9,10,11]. However, *Lactobacillus* spp. strains are limited by insufficient hydrolytic enzyme secretion, restricting their capacity to enhance substrate digestibility [11]. Although *Bacillus* spp. demonstrate robust fermentation activity, they generate odorous metabolites (e.g., indole and mercaptans), impairing feed palatability [12]. During fermentation processes, yeast strains biosynthesize an array of bioactive compounds including secondary metabolites, hydrolytic enzymes, organic acids, and essential amino acids, which collectively enhance the nutritional matrix of fermentation substrates through synergistic biochemical pathways [13]. Therefore, this study selected *Candida utilis* CU-3 as the fermentation strain for cottonseed hull processing. Lv et al. showed *Saccharomyces cerevisiae* LXPSC1 improved fermented meat quality through enhanced sensory profiles and elevated key biochemical parameters (pH, ethanol, FAA, VOCs) [14]. Vlassa et al. demonstrated that 24 h *Saccharomyces* fermentation reduced anti-nutritional factors (glucosinolates, 3-butyl isothiocyanate) and polyphenol-associated DPPH antioxidant activity [15].

Microbial consortia and their metabolic byproducts constitute pivotal determinants of fermented feed quality [16]. High-throughput sequencing and metabolomics have emerged as powerful tools to elucidate microbial community dynamics and metabolite signatures in fermented substrates [17]. Metabolomics comprehensively uncovers the metabolic profiles of all small-molecule metabolites in biological samples, aiming to perform qualitative and relative quantitative analysis of all metabolites [18]. Deng et al. revealed distinct microbial communities and volatile metabolite profiles between oat fermentation groups, with esters and terpenoids driving flavor distinctions in sweetness and fruity aroma, while *Lactobacillus* and *Enterococcus* emerged as dominant fermentative species [19]. Pynhunlang et al. revealed *Bacillus subtilis* as the dominant genus in Kinema through multi-omics integration, identifying anticancer (chrysin), antibacterial (benzimidazoles), and anti-HIV (3-hydroxy-L-kynurenine) bioactive metabolites [20]. By integrating multi-omics datasets of microbial consortia and their metabolic profiles, these approaches enable comprehensive characterization of community structural composition, functional annotation of metabolic pathways, and systemic delineation of microbiome-associated metabolic potential.

Currently, there are few reports on fermentation feed using cottonseed hulls as the substrate; there is a lack of relevant research on the nutritional components, microbial community changes, and metabolites of fermented cottonseed hulls. This study utilized *Candida utilis* CU-3, a strain with gossypol-degrading capability, to conduct solid-state fermentation of cottonseed hull. The effects of this fermentation on the nutritional composition of cottonseed were investigated. Furthermore, through microbiomics and metabolomics, the study evaluated the potential advantages of *Candida utilis* CU-3 fermented cottonseed hull feed, thereby laying a foundation for the development of fermented feed using cottonseed hull as a substrate.

## 2. Materials and Methods

### 2.1. Materials

The cottonseed hulls are sourced from Xuze Biotechnology Co., Ltd. (Changji, China).

The anaerobic solid-state fermentation bags were purchased from Bofeide Technology Co., Ltd. (Dongying, China).

*Candida utilis* CU-3 (preservation number: CGMCC No. 9165) was deposited at the China General Microbiological Culture Collection Center (CGMCC, Beijing, China).

### 2.2. Experimental Design and Sampling

*Candida utilis* CU-3 was inoculated into YPD medium and cultured in a shaking incubator at 30 °C and 150 rpm for 24 h to achieve optimal fermentation performance.

The fermentation substrate comprised cottonseed hull (100 g). The experimental group was inoculated with 5% (*v*/*v*) *Candida utilis* CU-3 culture (1.0 × 10^8^ CFU/mL), followed by supplementation with sterile water to achieve a moisture content of 60%. The control group received sterile water only to reach the same moisture level (60%). Both groups were transferred into anaerobic fermentation bags and incubated at 30 °C for 10 days, with fermentation samples collected at 5-day intervals.

### 2.3. Detection of Nutrient Composition

Cottonseed hull raw material and cottonseed hull-fermented feed were air-dried naturally, ground using a miniature plant sample grinder, passed through a 1 mm sieve, packed into zip-lock bags, and stored at 4 °C for subsequent analysis.

The determination of crude protein (CP), ether extract (EE), and crude fiber (CF) contents in the samples was performed according to the method described by Thiex [21].

### 2.4. Free Gossypol Content Determination

The free gossypol contents in both the experimental and control groups were determined in accordance with the international standard ISO 6866:1985 (Animal feeding stuffs—Determination of free and total gossypol) [22]. Free gossypol was extracted using a mixed solvent system of isopropanol and n-hexane in the presence of 3-amino-1-propanol, followed by conversion of gossypol to aniline–gossypol complex through a reaction with aniline. The absorbance of the resulting complex was measured colorimetrically at its maximum absorption wavelength of 440 nm.

### 2.5. ITS Sequencing

Total genome DNA from samples was extracted using the CTAB method. DNA concentration and purity were monitored on 1% agarose gel. According to the concentration, DNA was diluted to 1 ng/µL using sterile water. The ITS gene (ITS1 region) was amplified with specific primers ITS5-1737F and ITS2-2043R. PCR amplification was performed using Phusion^®^ High-Fidelity Master Mix (New England Biolabs) with the following: a 15 μL reaction volume (2 μm primers; 10 ng DNA template); a cycling protocol of 98 °C/1 min → 30 × [98 °C/10 s → 50 °C/30 s → 72 °C/30 s]; purification with Qiagen Gel Extraction Kit (Qiagen, Hilden, Germany); library construction using TruSeq^®^ DNA PCR-Free Kit (Illumina, San Diego, CA, USA) with dual-indexing per protocol; and quality controlled using Qubit^®^ 2.0 (Thermo Fisher, Waltham, MA, USA) and Bioanalyzer 2100 (Agilent, Palo Alto, CA, USA). Sequencing was performed using the Illumina NovaSeq (PE250) platform with 250 bp paired-end reads.

Demultiplex the sample data from the sequencing data based on barcode and PCR primer sequences, trim off these sequences, and then use FLASH (Version 1.2.7) to merge the reads, obtaining the raw Tags data. Subsequently, trim reverse primer sequences using Cutadapt to prevent interference with downstream analyses. Perform quality filtering on the Raw Tags with fastp to obtain high-quality Clean Tags. Finally, remove chimeric sequences by aligning against the SILVA or UNITE databases to generate the final Effective Tags.

Process the Effective Tags through quality control, trimming, and merging using QIIME2 (Version 1.9.1), followed by taxonomic annotation against the SILVA 138.1 reference database within the QIIME2 (Version 1.9.1) framework. Microbial community structural shifts in cottonseed hull feed after fermentation were elucidated via statistical analyses, including PCoA, relative abundance bar plots, species abundance clustering heatmaps, and LEfSe (LED > 3.0).

### 2.6. Untargeted Metabolomics

Thaw a 100 mg sample on ice and transfer it to an EP tube. Add 500 μL of 80% (*v*/*v*) aqueous methanol solution. Vortex thoroughly and incubate in an ice bath for 5 min. Centrifuge at 12,000 rpm and 4 °C for 20 min. Dilute a measured volume of the supernatant with MS-grade water to reduce methanol content to 53%. Re-centrifuge under the same conditions (12,000 rpm, 4 °C, 20 min) and collect the supernatant for LC-MS analysis.

Annotate metabolites using the KEGG (Kyoto Encyclopedia of Genes and Genomes), HMDB (Human Metabolome Database), and LIPID MAPS (LIPID Metabolites and Pathways Strategy) databases. Perform principal component analysis (PCA) and Partial Least Squares-discriminant Analysis (PLS-DA) using metaX software (version 1.4.2). Calculate Variable Importance in Projection (VIP) scores for metabolites. Compute Fold Change (FC) values for metabolites. Screening criteria for differential metabolites: VIP > 1, *p* < 0.05 (Student’s *t*-test or equivalent), FC ≥ 2 (upregulated) or FC ≤ 0.5 (downregulated). Volcano plots are generated using the ggplot2 R package (version 3.4.3), integrating VIP values, log_2_ (Fold Change), and −log_10_ (*p*-value) to identify significant metabolites. Calculate Pearson’s correlation coefficients using the cor function in R. Assess statistical significance via the cor.mtest method (*p* < 0.05). Visualize correlation matrices using the corrplot R package (version 3.4.3).

### 2.7. Statistical Analysis

Data are presented as mean ± standard deviation (n = 3 for nutritional analysis, n = 6 for microbial analysis and metabolic analyses). Statistical analysis was performed using SPSS 22.0 software (IBM Corporation, Armonk, NY, USA). Significant differences between experimental and control groups were determined by Student’s *t*-test and one-way analysis of variance (ANOVA), with a *p*-value < 0.05 considered statistically significant.

## 3. Results

### 3.1. Analysis of Feed Nutritional Composition and Free Gossypol Content

To ensure the nutritional value of fermented feed, we conducted determinations of crude protein, crude fat, crude fiber, and free gossypol in both control and experimental groups of fermented feed at different time points. As shown in Table 1, the crude protein (CP) and crude fat (EE) contents in the control group showed no significant differences (*p* > 0.05) with increasing fermentation time. The crude fiber (CF) content in the control group decreased significantly (*p* < 0.05) as fermentation progressed, reaching 36.95 ± 0.45% after 10 days of fermentation. In contrast, the free gossypol content in the control group exhibited significant variations (*p* < 0.05) during fermentation, with a measured value of 170.08 ± 2.23 mg/kg following 10 days of fermentation. In the experimental group, the crude protein (CP) content significantly increased with prolonged fermentation time (*p* < 0.05). However, no significant difference (*p* > 0.05) was observed between the 5-day and 10-day fermentation periods, although a progressive increase in crude protein (CP) content was still noted. The crude fat (EE) content showed no statistically significant variation (*p* > 0.05) during fermentation. Concurrently, the crude fiber (CF) content decreased significantly (*p* < 0.05) over time, reaching 31.61 ± 1.66% of crude fiber content after 10 days. Similarly, the free gossypol content exhibited significant time-dependent changes (*p* < 0.05), declining to 64.80 ± 4.35 mg/kg following 10 days of fermentation, which is below the national standard limit of 100 mg/kg. Compared with the control group, the treatment group demonstrated a statistically significant increase in crude protein (CP) content (*p* < 0.05), along with significant reductions in crude fiber (CF) content (*p* < 0.05) and free gossypol concentration (*p* < 0.05). No significant difference was observed in crude fat (EE) content between the two groups.

### 3.2. Microbial Diversity Analysis of Fermented Feed

To investigate the differences in microbial communities between experimental groups (CU0, CU5, CU10) and control groups (CK0, CK5, CK10), ITS sequencing was performed on fermented feed. The numbers 0, 5, and 10 represent the fermentation duration in days. The PCA-based clustering separated microbial communities into two groups: the control group (CK0, CK5, CK10) and experimental groups, with CU0 clustered with the control group and CU5/CU10 forming a distinct cluster. Fungal communities in the control group and experimental groups CU0 exhibited higher similarity, while those in the experimental groups (CU5 and CU10) demonstrated stronger similarity within their respective groups (Figure 1). Bar plot analysis at the genus level demonstrated that the dominant fungal composition shifted from *Biploaris* and *Aspergillus* in the control group to *Cyberlindnera* and other yeasts in the experimental group (Figure 2). Cluster heatmap analysis of genus-level fungal abundance revealed distinct compositional patterns between groups. The control and CU0 groups were predominantly composed of pathogenic fungi (e.g., *Bipolaris*, *Trichothecium*, and *Scedosporium*) and molds (e.g., *Mucor*, *Trichocladium*). In contrast, by day 5 and 10, the experimental groups (CU5 and CU10) showed a marked shift toward yeast-dominated communities (e.g., *Candida*, *Cyberlindnera*) (Figure 3). LEfSe analysis identified significant differences in fungal composition from the order to genus level between control and experimental fermented feed samples. Specifically, yeast taxa (e.g., *Saccharomyces*, *Cyberlindnera*) were significantly enriched in the experimental group, while the control group and experimental group exhibited higher abundances of pathogenic and saprophytic fungi, including *Biplaris*, *Rhizopus arrhizus*, *Pleosporaceae*, and *Dothideomycetes* (Figure 4).

### 3.3. Cluster Analysis of Metabolomics Between Experimental Group CU10 and Control Group CK10

To investigate metabolic changes in fermented feed, LC-MS/MS-based untargeted metabolomics analysis was conducted. We compared feed samples from two groups: the experimental group CU10 and the control group CK10. This approach detected 1313 metabolites through combined positive/negative ion mode acquisition. The principal component analysis (PCA) revealed significant differences in metabolite profiles between the experimental group CU10 and the control group CK10, as demonstrated in Figure 5a,d. The PLS-DA analysis of feed metabolites revealed distinct metabolic profile alterations, with the experimental group CU10 and control group CK10 clearly separated (Figure 5b,e). The PLS-DA models established for the experimental and control groups demonstrated robust validity, with model evaluation parameters obtained through 7-fold cross-validation. The R2 values of all models exceeded their corresponding Q2 values, and the Q2 regression line intercept with the Y-axis was below zero (Figure 5c,f). Collectively, these findings indicate model reliability without overfitting.

### 3.4. Identification of Differential Metabolites Between Experimental Group CK10 and Control Group CU10

The screening of differential metabolites relied on three parameters: VIP (Variable Importance in Projection), FC (Fold Change), and *p*-value. The thresholds were set as VIP > 1.0, |log2 FC| > 0.263, and *p*-value < 0.05. VIP values were derived from the first principal component of the PLS-DA (Partial Least Squares-discriminant Analysis) model, representing Variable Importance in Projection. FC indicates the Fold Change ratio between comparison groups, while *p*-value reflects the statistical significance of differences. The identified differential metabolites meeting these criteria are listed in Appendix A. The differential metabolites were classified into distinct categories based on ion mode. In the positive ion mode, these metabolites were categorized into phenylpropanoids and polyketides; amino acids and organic acids, along with their derivatives; and peptides and analogs, amino acids, and lipids and lipid-like molecules. In the negative ion mode, the differential metabolites comprised nucleosides, nucleotides, and analogs, phenylpropanoids and polyketides, amino acids and organic acids with their derivatives, and lipids and lipid-like molecules (Appendix A). Volcano plots visually illustrate the overall distribution of differentially expressed metabolites. Under positive ion mode, the differential metabolite volcano plot revealed that 33 metabolites were upregulated and 19 metabolites were downregulated (Figure 6a). Similarly, under negative ion mode, the volcano plot showed 19 upregulated metabolites and 22 downregulated metabolites (Figure 6b).

### 3.5. Heatmap Analysis of Differential Metabolites Between Experimental Group CK10 and Control Group CU10

The clustering heatmap analysis of differential metabolites visually demonstrates the distinct clustering patterns between CK10 and CU10, with rows representing clustered differential metabolites and columns corresponding to sample information. Color intensity reflects metabolite abundance, where red indicates upregulated metabolites and blue denotes downregulated metabolites (Figure 7). In the positive ion mode clustering analysis, metabolites significantly upregulated in CU10 included Sclareolide, Sedanolide, sorbic acid, L-saccharopine, 2-aminopimelic acid, chrysin, myricetin, and sclareol; in CK10, marked increases were observed in genipin, LPE 18:2, and 7-methylxanthine (Figure 7a). In the negative ion mode clustering analysis, metabolites significantly upregulated in CU10 included L-ornithine, 2-isopropylmalic acid, aucubin, porphobilinogen, oxyresveratrol, and thiamine, while CK10 exhibited marked increases in metabolites such as tetradecanedioic acid, N-acetylglycine, LPE 18:1, and genipin (Figure 7b).

### 3.6. KEGG Pathway Analysis of Differential Metabolites Between Experimental Group CK10 and Control Group CU10

Based on the metabolite classification information provided by the KEGG database, statistical visualization of annotated differential metabolites was conducted. The results demonstrated that in both positive and negative ion modes, these differential metabolites were distributed across three major categories: environmental information processing, genetic information processing, and metabolism (Figure 8). Under positive ion mode, 6 metabolites were involved in environmental information processing, 5 in genetic information processing, and 216 in metabolism. Among these, the metabolic pathways including membrane transport, translation, other amino acid metabolism, cofactor and amino acid metabolism, lipid metabolism, global and overview maps, carbohydrate metabolism, biosynthesis of other secondary metabolites, and amino acid metabolic pathways each contained more than 5 metabolites (Figure 8a). Under negative ion mode, 14 metabolites were involved in environmental information processing, 4 metabolites participated in genetic information processing, and 225 metabolites were engaged in metabolism. Among these, the metabolic pathways, including signal transduction, membrane transport, nucleotide metabolism, metabolism of terpenoids and polyketides, other amino acid metabolism, cofactor and amino acid metabolism, lipid metabolism, global and overview maps, energy metabolism, carbohydrate metabolism, biosynthesis of other secondary metabolites, and amino acid metabolic pathways, each contained more than five metabolites (Figure 8b).

Enrichment analysis of differential metabolites based on KEGG annotations was performed using the hypergeometric test implemented in the clusterProfiler package (Figure 9). Under positive ion mode, the enrichment results were primarily associated with metabolic pathways including metabolic pathways, flavonoid biosynthesis, isoflavonoid biosynthesis, vitamin B6 metabolism, lysine degradation, lysine biosynthesis, glycine, serine, threonine metabolism, and lipid metabolism (Figure 9a). Under negative ion mode, the enrichment results were predominantly linked to metabolic pathways including metabolic pathways, biosynthesis of secondary metabolites, glutathione metabolism, 2-oxocarboxylic acid metabolism, valine, leucine, and isoleucine biosynthesis, riboflavin metabolism, and pyruvate metabolism (Figure 9b).

## 4. Discussion

Cottonseed hulls face utilization constraints as roughage feed, particularly in non-ruminant nutrition, due to free gossypol toxicity [23]. Thus, numerous studies have been dedicated to developing effective methods to reduce free gossypol content in cottonseed byproducts, thereby enabling their safe utilization in animal feed [24,25]. Among various strategies, solid-state microbial fermentation stands out as one of the most effective approaches. This process can reduce levels of anti-nutritional factors, produce various functional enzymes, and enhance the nutritional content of fermented substrates through microbial metabolic activities [6]. In this study, cottonseed hulls processed without autoclaving, crushing, or screening demonstrated significant improvements following solid-state fermentation compared to the control group. The crude protein content increased by 27.78% (*p* < 0.05), crude fiber content decreased by 5.07% (*p* < 0.05), and free gossypol levels were reduced by 61.90% (*p* < 0.01). No significant variation was observed in crude fat content (*p* > 0.05). This indicates that during the fermentation process, yeast can convert nitrogen sources in feed into proteins stored within their cells, thereby increasing the crude protein content of roughage. Simultaneously, yeast secretes hydrolytic enzymes capable of degrading various nutritional components and anti-nutritional factors, consequently enhancing the nutritional value of cottonseed hulls [13,26,27]. Niu et al. showed that solid-state fermentation of cottonseed meal by *Candida tropicalis* increased crude protein content and reduced free gossypol levels, which is consistent with our results [28].

Microbial analysis revealed that solid-state fermentation with *Candida utilis* CU-3 inoculation significantly altered the microbial community dynamics of the fermented feed substrate. In the experimental group, *Cyberlindnera* progressively increased in relative abundance at the genus level as fermentation duration extended, eventually becoming the dominant genus. Concomitantly, the relative abundances of *Bipolaris* and *Aspergillus* at the genus level decreased significantly in proportion to fermentation time. This inhibitory effect may be attributed to certain organic compounds produced by *Candida utilis* CU-3 during anaerobic fermentation, consistent with the findings of Zhang et al. [29]. Simultaneously, changes in microbial communities may serve as the primary driver of metabolite variation. He et al. [30] found that shifts in metabolites correlated with microbial community dynamics across different ensiling stages of soybean, which aligns with the observations in this study.

Untargeted metabolomics data revealed that among differential metabolites, phenylpropanoids and polyketides were predominantly involved in the biosynthesis of flavones and flavonoids. The predominant flavones and flavonoids identified in fermented cottonseed hulls, including chrysin and myricetin, demonstrated multifunctional pharmacological activities encompassing antioxidant, anti-inflammatory, and immunomodulatory properties [31,32]. Kang et al. demonstrated that myricetin increased the abundance of butyrate-producing bacteria and reduced plasma lipopolysaccharide (LPS) levels, thereby alleviating inflammatory responses through inhibition of TLR4 activation [33]. Gong et al. showed that chrysin exhibited a potent anti-porcine epidemic diarrhea virus (PEDV) activity by enhancing cellular viability and reducing viral copy numbers [34]. Additionally, chrysin demonstrated a favorable safety profile in animals [35]. Based on the aforementioned findings, the cottonseed hull inoculated with *Candida utilis* CU-3 exhibited a significant increase in flavone and flavonoid compound content. This enhancement may confer antioxidant, antibacterial, and anti-inflammatory properties to the fermented cottonseed hull, thereby playing a crucial role in improving cottonseed hull fermented feed through pathogen inhibition, animal stress reduction, and immune function enhancement.

Among the differential metabolites, significantly increased amino acids and organic acids/their derivatives included L-ornithine, 2-isopropylmalic acid, and 4-methoxycinnamic acid. These findings are consistent with the enriched metabolic pathways identified in the KEGG analysis, specifically those related to amino acid and organic acid biosynthesis. These amino acids and organic acid derivatives likely play critical roles in *Candida utilis* CU-3 fermented cottonseed hulls, potentially promoting animal growth and development. L-ornithine serves as a critical intermediate in the urea cycle, converting toxic ammonia generated from protein metabolism in animals into urea for excretion [36]. This biochemical process effectively mitigates the risk of ammonia toxicity while protecting hepatic function, as demonstrated by studies showing that ornithine supplementation enhances ammonia detoxification efficiency and mitigates the effects of uremic toxins in cats [37]. In previous studies, 2-isopropylmalic acid was synthesized in yeast and demonstrated moderate antioxidant and antimicrobial activities [38]. Therefore, 2-isopropylmalic acid may indirectly enhance the protein content in feed or optimize the amino acid composition. In previous studies, 4-methoxycinnamic acid has demonstrated antibacterial activity, amelioration of abnormal glucose metabolism, and neuroprotective effects [39,40]. The inoculation of *Candida utilis* CU-3 significantly increased amino acids, organic acids, and their derivatives in cottonseed hulls. This enhancement may confer functional properties to fermented hulls, including improved microbial metabolism and protein utilization efficiency. Consequently, fermented cottonseed hull feed may play a crucial role in improving feed quality and promoting animal health through these mechanisms.

Among the differential metabolites, the significantly increased organic heterocyclic compounds included Sclareolide, Sedanolide, and others. This observation aligns with the enriched metabolic pathways identified in the KEGG analysis, particularly those related to secondary metabolite biosynthesis. The majority of these compounds are classified as intermediate pharmaceutical molecular structures, exhibiting strong structural correlations with gossypol degradation products, particularly when gossypol undergoes enzymatic hydrolysis or sequential metabolic pathways to be degraded into low-molecular-weight ketones and esters [41]. Sclareolide, a sesquiterpene lactone organic compound, demonstrates multifaceted biological activities including antibacterial, anticancer, anti-inflammatory, and cytotoxic effects [42]. Chen et al. [43] discovered that Sclareolide exhibits anti-Ebola virus (EBOV) activity, functioning as an EBOV fusion inhibitor to suppress the replication of eight filoviruses. Sedanolide, a bioactive compound exhibiting antibacterial and antitumor activities, is primarily derived from celery oil [44]. Yosuke et al.’s study revealed that Sedanolide activates the KEAP1-NRF2 signaling pathway, thereby enhancing cellular resistance to oxidative damage [45]. This indicates that the inoculation of *Candida utilis* CU-3 significantly increased the content of organic heterocyclic compounds in cottonseed hulls, potentially endowing the fermented cottonseed hulls with functional properties such as bacterial growth inhibition and feed spoilage delay. Consequently, fermented cottonseed hull feed may play a crucial role in enhancing antimicrobial efficacy and extending feed preservation through these mechanisms.

Furthermore, among the differentially abundant metabolites, significantly increased compounds included (20R)Ginsenoside Rh2, Pseudoginsenoside RT5, and 4-methoxybenzaldehyde. (20R)Ginsenoside Rh2 and Pseudoginsenoside RT5, both naturally bioactive compounds primarily derived from *Panax ginseng*, exhibit neuroprotective, cardioprotective, antimicrobial, immunomodulatory, and anticancer activities [46]. These pharmacological properties may contribute to functional feed development through their bioactive mechanisms. Chen et al. demonstrated that (20R)Ginsenoside Rh2 exhibits significant antidepressant effects in murine models [46]. 4-Methoxybenzaldehyde, a volatile organic compound (VOC) with an anise-like odor [47], may enhance feed palatability by modulating aromatic profiles, thereby stimulating feeding behavior in animals. Overall, these findings suggest that *Candida utilis* CU-3 inoculated cottonseed hulls may confer functional advantages to fermented cottonseed hull feed, particularly in enhancing immune activity and promoting feed intake.

## 5. Conclusions

In this study, solid-state fermentation of cottonseed hulls by *Candida utilis* CU-3 significantly enhanced the crude protein content while reducing crude fiber and free gossypol levels. From the perspective of process economic feasibility analysis, cottonseed hulls—as a by-product of cotton processing—demonstrate extensive availability and low-cost advantages. The solid-state fermentation process requires no complex equipment investment and can be conducted in conventional fermentation vessels, characterized by relatively low energy consumption and straightforward operational procedures. Microbiome analysis and metabolomics revealed that *Candida utilis* CU-3 became the dominant microbial during the fermentation process, concurrently synthesizing functional bioactive metabolites such as chrysin, myricetin (with anti-inflammatory, antibacterial, and antioxidant properties), and Ginsenoside Rh2 (exhibiting anticancer, antimicrobial, and neuroprotective activities). Therefore, this study demonstrates that cottonseed hulls inoculated with *Candida utilis* CU-3 serve as a rich source of diverse bioactive compounds, immunomodulators, and vitamins, providing a theoretical foundation for developing cottonseed hull-based fermented feed.

## Figures and Tables

**Figure 1 microorganisms-13-01380-f001:**
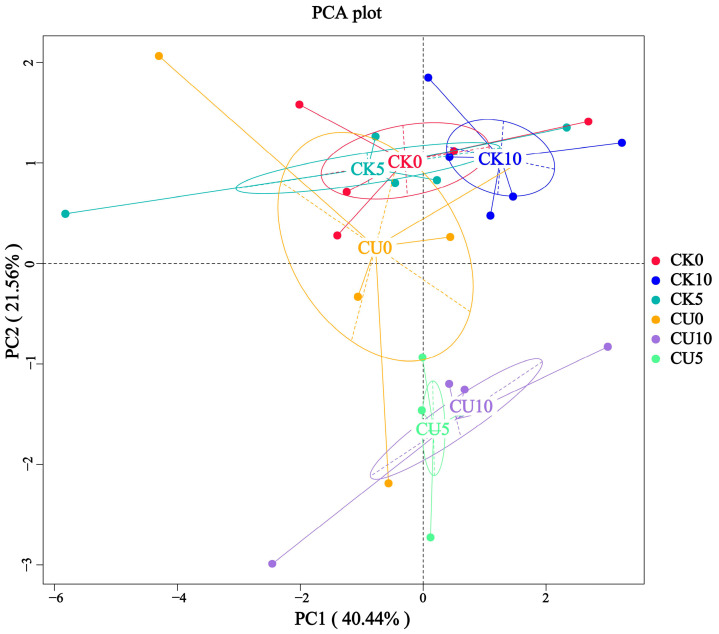
Principal coordinate analysis (PCA) plot of fungal diversity in feed samples from control and experimental groups, demonstrating distinct clustering patterns.

**Figure 2 microorganisms-13-01380-f002:**
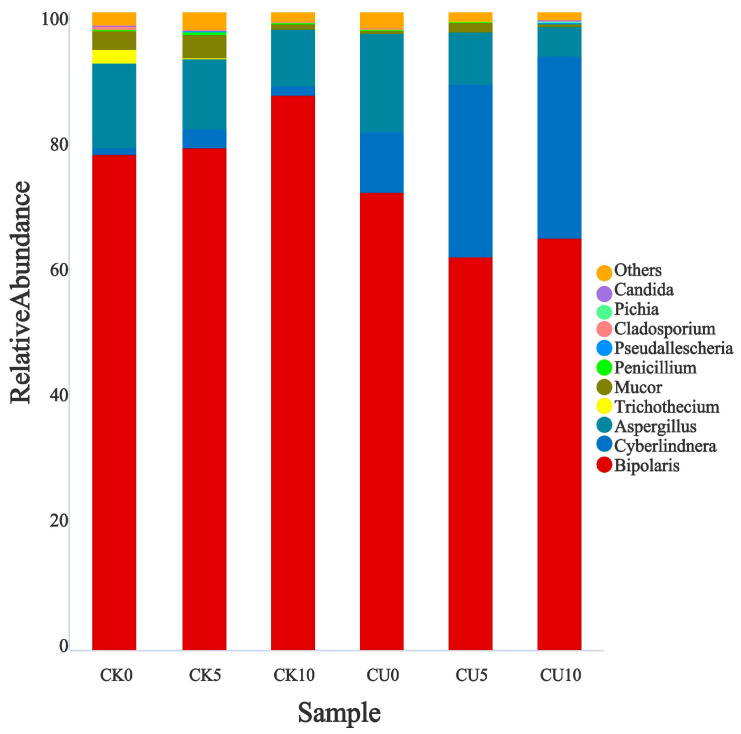
Relative abundance of key fungal genera in feed samples from control and experimental groups.

**Figure 3 microorganisms-13-01380-f003:**
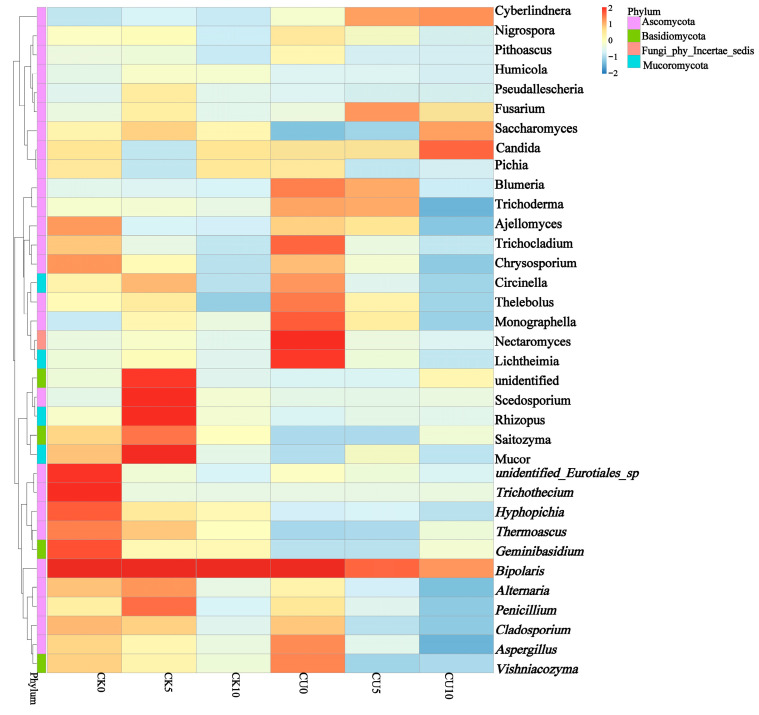
Clustering heatmap of species abundance at the genus level in feed samples from control and experimental groups, based on hierarchical clustering analysis.

**Figure 4 microorganisms-13-01380-f004:**
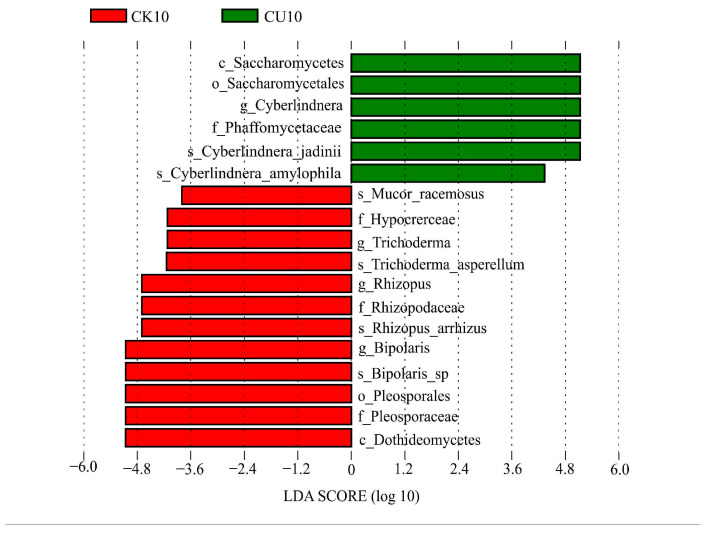
Significantly differential features identified by LEfSe (Linear Discriminant Analysis Effect Size) between experimental group CU10 and control group CK10, with an LDA score threshold > 3.0.

**Figure 5 microorganisms-13-01380-f005:**
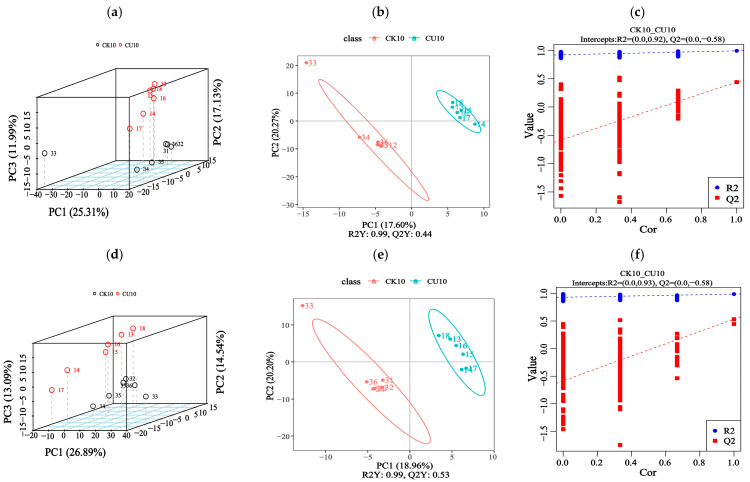
Metabolomic analysis of fermented feed using PCA and PLS-DA (positive/negative ion modes). (**a**–**c**) Positive ion mode analyses; (**d**–**f**) negative ion mode analyses. (**a**,**d**) Principal component analysis (PCA) score plots under positive and negative ion modes, respectively. (**b**,**c**) Partial Least Squares-discriminant Analysis (PLS-DA) of differential metabolites in positive ion mode, displaying score plots (**b**) and overview of PLS-DA Model Validation Plots (**c**). (**e**,**f**) PLS-DA of differential metabolites in negative ion mode, showing score plots (**e**) and overview of PLS-DA Model Validation Plots (**f**).

**Figure 6 microorganisms-13-01380-f006:**
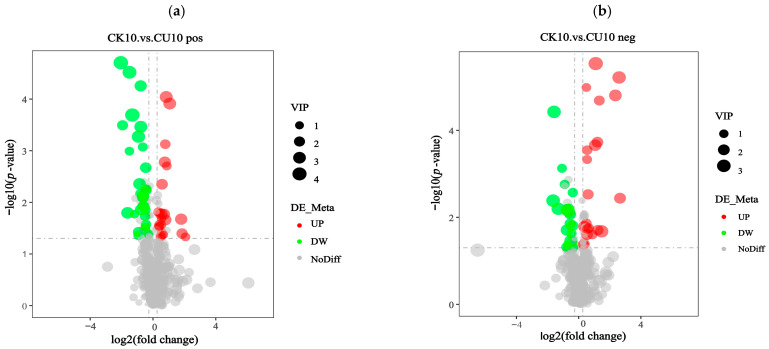
Volcano plots of differential metabolites in positive/negative ion modes. Fold Change (log_2_ [FC]) is plotted on the x-axis and statistical significance (−log_10_ [*p*-value]) is on the y-axis. Upregulated metabolites (UP) and downregulated metabolites (DW) are represented by red and green circles, respectively. The size of the circles corresponds to the Variable Importance in Projection (VIP) scores, reflecting the significance of metabolites in the projection model. (**a**) Volcano plot of differential metabolites in positive ion mode. (**b**) Volcano plot of differential metabolites in negative ion mode.

**Figure 7 microorganisms-13-01380-f007:**
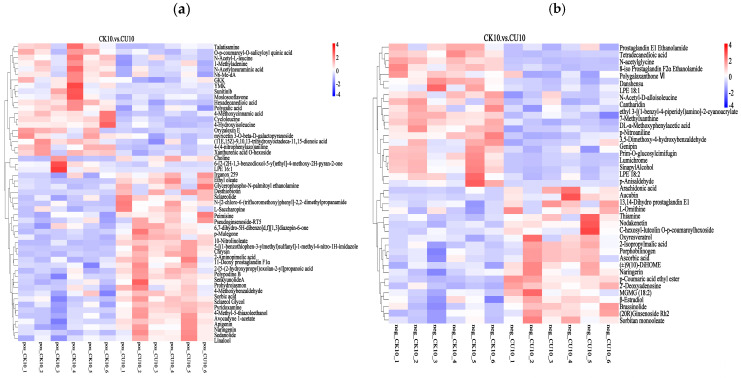
Clustering heatmap of differential metabolites, with columns representing grouped differential metabolites and rows indicating sample information. (**a**) Hierarchical clustering analysis of differential metabolites in positive ion mode. (**b**) Hierarchical clustering analysis of differential metabolites in negative ion mode.

**Figure 8 microorganisms-13-01380-f008:**
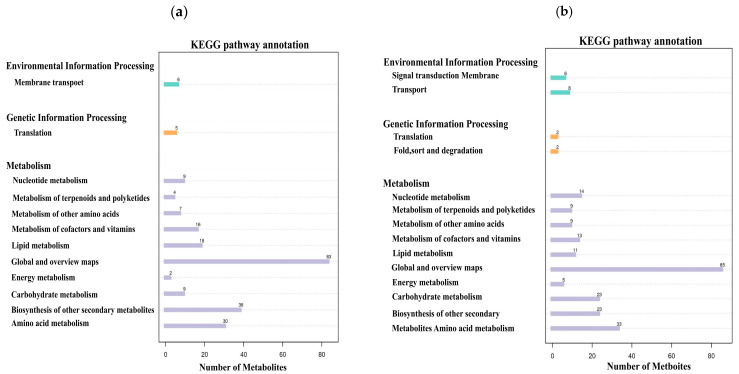
KEGG pathway annotation bar plots, with the x-axis representing metabolite count and the y-axis indicating metabolic pathways. (**a**) KEGG pathway annotation of differential metabolites in positive ion mode. (**b**) KEGG pathway annotation of differential metabolites in negative ion mode.

**Figure 9 microorganisms-13-01380-f009:**
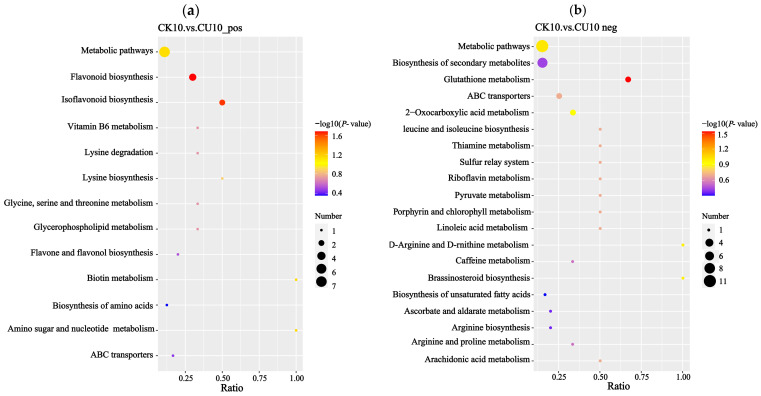
KEGG enrichment analysis bar plots, with the x-axis representing the enrichment ratio (metabolite count ratio) and the y-axis indicating metabolic pathways. (**a**) KEGG enrichment results of differential metabolites in positive ion mode. (**b**) KEGG enrichment results of differential metabolites in negative ion mode.

**Table 1 microorganisms-13-01380-t001:** The difference in nutritional analysis of CK10 between the experimental group CU10 and the control group CK10.

Items	Fermentation Time	Control Group	Treatment Group
Crude protein (CP, %)	0d	2.66 ± 0.10 ^Aa^	2.68 ± 0.14 ^Ab^
	5d	2.76 ± 0.13 ^Ba^	3.52 ± 0.28 ^Aa^
	10d	2.88 ± 0.09 ^Ba^	3.68 ± 0.27 ^Aa^
Crude fat (EE, %)	0d	3.23 ± 0.15 ^Aa^	3.37 ± 0.15 ^Aa^
	5d	3.43 ± 0.32 ^Aa^	3.50 ± 0.20 ^Aa^
	10d	3.60 ± 0.10 ^Aa^	3.30 ± 0.20 ^Aa^
Crude fiber (CF, %)	0d	38.70 ± 0.93 ^Aa^	37.97 ± 0.25 ^Aa^
	5d	38.24 ± 0.51 ^Aab^	36.30 ± 0.94 ^Ba^
	10d	36.95 ± 0.45 ^Ab^	31.61 ± 1.66 ^Bb^
Free gossypol (FG, mg/kg)	0d	189.25 ± 2.31 ^Aa^	186.95 ± 4.32 ^Aa^
	5d	188.97 ± 1.66 ^Aa^	129.60 ± 8.59 ^Bb^
	10d	170.08 ± 2.23 ^Ab^	64.80 ± 4.35 ^Bc^

Different superscript uppercase letters within the same row (e.g., A, B) indicate significant between-group differences (*p* < 0.05), while different superscript lowercase letters within the same column (e.g., a, b) denote significant within-group differences (*p* < 0.05).

## Data Availability

The original contributions presented in this study are included in the article. Further inquiries can be directed to the corresponding authors.

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
