# Peer review of "Integrated Microbiota and Metabolomics Analysis of Candida utilis CU-3 Solid-State Fermentation Effects on Cottonseed Hull-Based Feed"

_microorganisms, 2025, doi:10.3390/microorganisms13061380_

Round 1
Reviewer 1 Report
Comments and Suggestions for Authors
This study investigates enhancing cottonseed hulls' nutritional value via anaerobic solid-state fermentation using Candida utilis CU-3, exploring the fermented feed's functional potential through fungal community dynamics and metabolite profiling. The results showed a 61.90% increase in free gossypol degradation, a 27.78% improvement in crude protein, and a 5.07% reduction in crude fiber. Untargeted metabolomics revealed functional bioactive compounds, including chrysin, myricetin, and ginsenoside Rh2, suggesting enhanced health-promoting potential.
Abstrac: Not comment.
Introduction: There is a lack of theory on metabolomics and some examples related to reactors of the same type.
Materials and Methods: Detection of nutrient composition. reference.
ITS sequencing. reference.
Results: Identification of Differential Metabolites Between Experimental Group CK10 and Control Group CU10. The volcano plot information is good, but if you performed PCA and 3dPCA, why don't you use that information to relate it to the metabolites? Review this data.
VIP analysis should generate the most important and highly correlated metabolites. You should use this information generated by PLS.
Figure 5 and 6. Low quality.
Discussion: Not comment.
Conclusions; The new requested data would need to be entered.
Author Response
Dear Reviewer,
Thank you very much for your careful review of our paper and the valuable comments you provided. In response to your feedback, we have made comprehensive and meticulous revisions to the manuscript. To facilitate your review of our modifications and detailed responses to each comment, we have submitted a document containing the reply as an attachment. In the attachment, each of your comments is clearly numbered, and we have provided a detailed response to each one, along with explanations of the corresponding modification locations and specific content in the paper.
Thank you again for your hard work and professional suggestions. We look forward to your further guidance on the revised version.
Sincerely,
Dong Deli
May 28, 2025

Reviewer 2 Report
Comments and Suggestions for Authors
Peer Review Report
Manuscript Title: Integrated Microbiota and Metabolomics Analysis of Candida utilis CU-3 Solid-State Fermentation Effects on Cottonseed Hull-Based Feed
Journal: Microorganisms
General Evaluation: The manuscript presents a well-structured and timely investigation into the application of Candida utilis CU-3 in the solid-state fermentation of cottonseed hulls. The integrated use of microbiota and metabolomics analysis is commendable and adds scientific rigor to the assessment of fermentation outcomes. The manuscript contributes to the advancement of knowledge in the valorization of agro-industrial byproducts through microbial biotechnology. However, some revisions are necessary to improve clarity, data visualization, and standardization.
Figure 1: This figure contains multiple sub-panels illustrating microbial community analyses, but the text within the genus/species names is not legible. It is essential to separate the sub-panels (a–d) into distinct figures or provide them at higher resolution. Additionally, microbial genus and species names must be italicized according to scientific standards.
Figure 4: The compound names in the metabolite clustering heatmaps are not clearly legible. The figure should also be separated or redesigned at a higher resolution, especially to facilitate interpretation of individual metabolites and their relative abundance.
Throughout the manuscript, ensure that all genus and species names are italicized (e.g., Candida utilis, Saccharomyces cerevisiae, Cyberlindnera, Trichothecium, etc.), especially in the text body, figure legends, and tables.
The methods are described in reasonable detail; however, the manuscript would benefit from specifying the software version and statistical thresholds used in the microbiota analysis (e.g., QIIME2 plugin versions, LEfSe LDA score cutoffs).
The metabolomic processing pipeline is rigorous; yet, please report the number of biological replicates per group for both sequencing and LC-MS/MS analyses.
Terminology: Be consistent with the naming of control and treatment groups (e.g., CK0 vs. CK-0 vs. CK 0). Unify the style throughout the manuscript and in the figure labels.
Spelling and Grammar: There are some minor typographical errors (e.g., "Curde protein" instead of "Crude protein") that should be corrected throughout the tables and text.
References: The citation style is consistent with MDPI formatting. However, ensure all DOIs are active and that references to supplementary methods (e.g., ISO standard for gossypol) are correctly formatted.
Comments on the Quality of English LanguageSpelling and Grammar: There are some minor typographical errors (e.g., "Curde protein" instead of "Crude protein") that should be corrected throughout the tables and text.
Author Response

(The authors gave the same response as above.)

Reviewer 3 Report
Comments and Suggestions for Authors
The manuscript reports on solid state fermentation of cottonseed hulls. This is an important and interesting subject in value added processing. Some comments to be addressed are as below:
1)Surprisingly, lines are not numbered that makes addressing comments difficult.
2)Abstract: Why an anaerobic fermentation was employed, while yeasts/molds are aerobic?
3)Under Materials and Methods, a section on ‘Materials’ with details is needed.
4)Introduction section; This should be ‘were investigated’ instead of ‘were investigate’.
5)Under Methods; why no air was used for fermentation while yeats are aerobic? ‘’Both groups were transferred into anaerobic fermentation bags and incubated at 30°C for 10 days, with fermentation samples collected at 5-day intervals.’’
6)How oxygen and CO2 levels were monitored during the process?
7)Section 2.3: Some more details on gossypol analysis should be presented.
8)Table 1: Please double check the (alphabetical) statistical ranking and do the corrections.
9)Conclusion: Please add brief information on economic feasibility of this process.
Author Response

(The authors gave the same response as above.)

Round 2
Reviewer 1 Report
Comments and Suggestions for Authors
Dear author, thank you for making all the suggested changes. I have no further comments on the work.
Reviewer 3 Report
Comments and Suggestions for Authors
The comments are addressed carefully and the manuscript is in good shape now. It is recommended for publication.